# Mental health is positively associated with biodiversity in Canadian cities
Rachel T. Buxton [1,9] ✉, Emma J. Hudgins[1,2,9], Eric Lavigne[3,4], Paul J. Villeneuve[5], Stephanie A. Prince [4,6], Amber L. Pearson [7], Tanya Halsall[8], Courtney Robichaud[1] & Joseph R. Bennett[1]

Cities concentrate problems that affect human well-being and biodiversity. Exploring the link between mental health and biodiversity can inform more holistic public health and urban planning. Here we examined associations between bird and tree species diversity estimates from eBird community science datasets and national forest inventories with self-rated mental health metrics from the Canadian Community Health Survey. We linked data across 36 Canadian Metropolitan Areas from 2007-2022 at a postal code level. After controlling for covariates, we found that bird and tree species diversity were significantly positively related to good self-reported mental health. Living in a postal code with bird diversity one standard deviation higher than the mean increased reporting of good mental health by 6.64%. Postal codes with tree species richness one standard deviation more than the mean increased reporting of good mental health by 5.36%. Our results suggest that supporting healthy urban ecosystems may also benefit human well-being.

Urban growth is a major contributor to biodiversity loss[1,2]. Yet, for approximately 55% of the global human population that live in cities, urban biodiversity represents the predominant exposure to nature[3]. Urban environments can contribute to stress[4] and limit opportunities for people to engage with the natural environment[5], which is problematic because exposure to nature is associated with a broad array of health and well-being benefits[6]. As urbanization increases, understanding the link between biodiversity and human health within an urban context will be a key component of effective and holistic urban conservation and public health planning now and into the future.

More than 50% of the population in middle- and high-income countries will suffer from at least one mental health disorder at some point in their lives[7]. In Canada, an estimated 1 in 5 people are affected by mental illness, costing upwards of 50 billion dollars annually and driving higher rates of disability and mortality[8,9]. As the population continues to age and urbanize, it is estimated that within a generation, 8.9 million Canadians will be living with a mental illness[8]. Neighborhood characteristics and geographic inequalities predominantly explain mental health outcomes, particularly the characteristics of the urban environment[10].

It is widely recognized that exposure to nature provides mental health benefits to people living in cities[11]. Two key psychological theories posit the

relationship between human health and wellbeing and natural environments: attention restoration theory (where nature facilitates the recovery from mental fatigue and replenishes attention through unconscious, cognitive processes[12]) and stress recovery theory (where natural environments facilitate the recovery from physiological stress through autonomic response[13]). Moreover, natural environments can reduce exposure to urban stressors, such as heat, noise, and air pollution[14] and promote physical activity[15,16] and social cohesion[17]. Much of this research considers the restorative properties of the amount of and proximity to greenspace, blue space, urban nature, and parks[18,19].

However, the role of biodiversity and different components of the natural environment in mental health research is unresolved[20]. Biodiversity is defined as the variety of life on Earth, from genes to ecosystems, associated with ecosystem functioning, resilience, and health[21]. Biodiversity is commonly measured in terms of species or taxonomic richness and diversity over a geographic area at a particular time[22]. The restorative effects of biodiversity are thought to relate to human evolution, throughout which our species has relied on a variety of species for survival and reproduction[23]. Thus, the Biophilia Hypothesis posits that humans have an innate affinity to connect with other species and nature (ref. 24, although for a critique of the Biophilia Hypothesis see ref. 25). Higher biodiversity may suggest a safe

[1]Department of Biology and Institute of Environmental and Interdisciplinary Sciences, Carleton University, Ottawa, ON, Canada. [2]School of Agriculture, Food, and Ecosystem Sciences, University of Melbourne, Parkville, VIC, Australia. [3]Environmental Health Science and Research Bureau, Health Canada, Ottawa, ON, Canada. [4]School of Epidemiology and Public Health, Faculty of Medicine, University of Ottawa, Ottawa, ON, Canada. [5]Department of Neuroscience, Carleton University, University, Ottawa, ON, Canada. [6]Centre for Surveillance and Applied Research, Public Health Agency of Canada, Ottawa, ON, Canada. [7]CS Mott Department of Public Health, Michigan State University, Flint, MI, USA. [8]University of Ottawa Institute of Mental Health Research at the Royal, Ottawa, ON, Canada. [9]These authors contributed equally: Rachel T. Buxton, Emma J. Hudgins. ✉e-mail: Rachel.buxton@carleton.ca

environment, where a variety of species mean that our needs are being met, leading to psychological restoration and perceived restorativeness[26]. However, how contact with biodiversity leads to better mental health and well-being outcomes is not well understood[27]. Studies have found beneficial relationships between the number and abundance of species and self-reported well-being, pleasure, connectedness to nature, overall health, stress, anxiety, and depression[26]. Within urban parks, improved psychological well-being was found to be related to higher bird and plant species richness[28], while lower depression, anxiety, and stress was associated with afternoon bird abundance[29]. In larger national studies, plant and bird species richness have been positively associated with mental health[30,31]. Conversely, other research has found a lack of association between standardized assessments of biodiversity and physiological well-being[32] or a positive association between well-being and perceived species diversity, but not for objective measures of diversity[33].

Associations between nature and health can vary between socio-economic status (SES) groups. Some studies have found the benefits of nature are greater in lower SES neighborhoods[34–36]. Much research has shown that low-income neighborhoods have reduced greenspace availability and biodiversity, termed the luxury effect[37]. Moreover, residents of lower SES are less likely to use greenspace that does exist[38]. However, inequality in biodiversity distribution can be driven instead by urban form, social policy, or collective human preference[39]. Studies of the relationship between health and biodiversity often occur in more affluent populations, precluding generalizable relationships between health, SES, and nature[40].

Given the recognition of the link between health and nature, and mounting environmental and mental health crises, nature-based solutions that address both of these issues are becoming increasingly widespread[41]. Nature-based solutions aim to protect, restore, and create biodiversity while addressing societal challenges[42]. For example, planting trees along streets can provide habitat for birds[43] while reducing the risk of depression[44], urban heat[45], and air pollution[46]. As interest in nature-based solutions increases, understanding the relationship between biodiversity and mental health can help guide national policy and planning.

Previous work has found that neighborhood greenness (measured using the Normalized Difference Vegetation Index, NDVI) is associated with lower odds of poor self-rated mental health in Canadian cities[47]. However, the role of biodiversity in this relationship is unknown. In general, because metrics of species diversity at large scales are rare, exploring the association between objectively measured biodiversity and mental health across a large number of cities is scarce. Here, we draw on the world's largest biodiversity-focused community (aka citizen) science platform, eBird, and a national forestry inventory to quantify bird and tree species diversity across Canadian urban areas. We explore the association between species diversity and mental health indicators, including self-rated mental health and stress from the Canadian Community Health Survey (CCHS), while accounting for the potential confounding effects of individual socio-demographic and health behavior characteristics. We also explore whether mental health indicators are better predicted by species diversity or other metrics of urban nature exposure, including NDVI and distance to and proportion of green and blue space. Finally, because of the importance of neighborhood-level SES[48], we explore whether the association between biodiversity and mental health varies based on area-level socioeconomic status. We hypothesize that bird and tree diversity will be associated with higher self-rated mental health and lower stress, that this association will be stronger than other metrics of nature, and this association will be particularly strong for lower income populations[31].

## Results

The CCHS is a repeated cross-sectional survey administered annually to collect detailed information on >1000 demographic and health-related variables. We used survey responses that assessed participants' self-rated mental health and stress[49] from respondents aged 18 and older from 2007 to 2021. Because CCHS health data were georeferenced at the six-digit postal code level, and other variables were aggregated to this level, tree and bird diversity metrics were aggregated by postal code and postal codes were used to link all data sources. To focus on the urban population of Canada, we restricted our analyses to postal codes in Canadian Census Metropolitan Areas (CMA, i.e., metropolitan areas with populations >100,000). To reduce within-postal code variability, very large postal codes with an area >16 km² (75th percentile of postal codes areas in CMAs) were excluded (n = 371,928 CCHS survey respondents). We also excluded postal codes with insufficient community science data to make accurate estimates of bird species diversity (see Materials and Methods, final sample size for self-rated mental health n = 47,623 and self-rated stress n = 48,693).

### Descriptive results

We created binary variables of good self-rated mental health and low self-rated stress using survey responses where participants ranked mental health as 'excellent' 'very good' or 'good' and most days as 'not at all stressful', 'not very stressful', and 'a bit stressful'. The proportion of participants who reported good mental health was 0.92 (standard deviation, SD = 0.27) and the proportion of participants who reported low stress was 0.80 (SD = 0.40, Supplementary Table S1).

### Sampling bias

In CMA postal codes containing sufficient data, CCHS respondents had a mean of $4800 higher income per year, were 1.94 years older, had 2.24 h more physical activity per week, and consumed 0.20 fewer alcoholic drinks per week than the overall set of CCHS respondents (Supplementary Table S1). This suggests an overrepresentation of wealthier, more active and older people who consume fewer alcoholic drinks within our sample. Conversely, in CMA postal codes containing sufficient data, CCHS respondents consumed a mean of 0.25 fewer servings of fruits and vegetables per day, and were exposed to 0.14 lower NDVI and 0.29 fewer tree species. These patterns largely persisted when data were stratified by neighborhood SES (i.e., marginalization, see Materials and Methods, Supplementary Table S1). There was very little difference in the probability of good self-rated mental health (0.92-0.93, low-high SES) and of low self-rated stress (0.78–0.81, low-high SES, Supplementary Table S1).

### Model results

We created a conceptual model to explore the relationship between mental health (dependent variables), species diversity and greenspace/bluespace (independent variables), and moderators and covariates (Supplementary Fig. S1). We used a generalized additive modeling (GAM) approach (details in Supplementary Methods) to explore the relationship between probability of poor self-rated mental health and high self-rated stress. To explore whether mental health indicators are better predicted by species diversity or other metrics of urban nature exposure, for each response variable we ran a model including species diversity and green and blue space-related variables. To explore how robust the results were to potential covariates we fit two other models: one where we added sociodemographic characteristics of survey respondents from the CCHS, and another where we added all potential covariates: sociodemographic characteristics and health behaviors. For each model set we then stratified data by low and high neighborhood SES.

We modeled the association with several different species diversity and greenness indices. The model with normalized difference vegetation index (a metric of 'greenness,' NDVI) measured at a 500 m buffer (see methods), tree species richness, modeled bird Shannon diversity, and year term linearized to minimize concurvity was the most parsimonious (lowest AIC = 26,185.90, deviance explained = 1.52%, n = 47,623, compared to models including other biodiversity and greenness variables Supplementary Table S2).

### Self-rated mental health

Shannon diversity of birds, an index of bird diversity taking into account the abundance of species, was significantly positively related to self-rated mental health in all models, regardless of covariates added and data stratification by

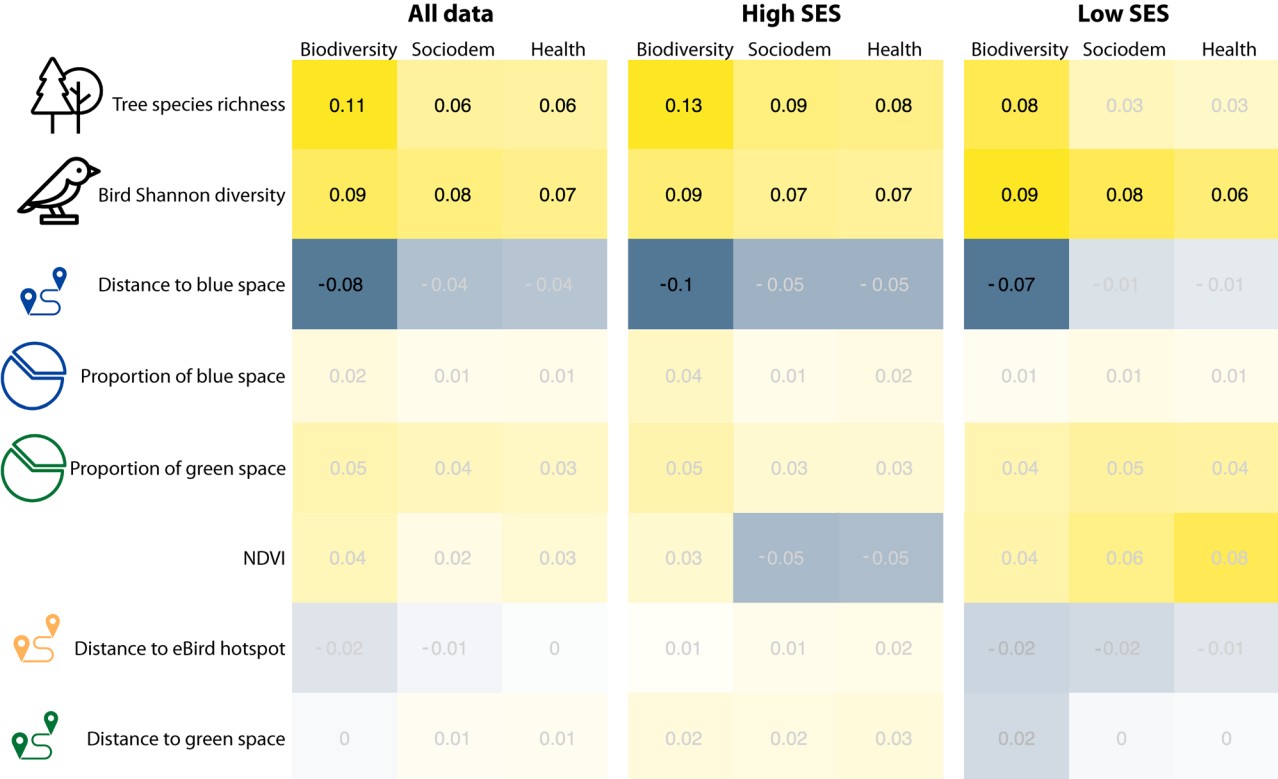

**Fig. 1 | Bird species diversity and tree species richness were significantly related to self-rated mental health.** Number represent parameter estimates of generalized additive models exploring the association between biodiversity variables and self-rated mental health from the Canadian Community Health Survey (CCHS). Grayed out values were not statistically significant (95% confidence intervals around parameter estimates overlapped with zero). In "biodiversity" models, only species diversity and greenspace and bluespace variables were included, "socio-dem" models added socio-demographic variables, and "health" models added health behavior and socio-demographic variables. All data were included, and data were stratified by neighborhoods with high socio-economic status (low marginalization) and low socio-economic status (high marginalization). Icons from the Noun Project.

SES (Fig. 1). Parameter estimates of models including data from only postal codes with low SES ($\beta_{birds}$ = 0.06, SE = 0.02, $p$ = 0.01) were similar and slightly lower than those including all data or data with only postal codes with high SES (all data: $\beta_{birds}$ = 0.07, SE = 0.02, $p$ < 0.001, high SES: $\beta_{birds}$ = 0.07, SE = 0.03, $p$ = 0.02).

Similarly, tree species richness was significantly positively associated with self-rated mental health in most models, except when data were subset to postal codes with low SES and demographic and health behavior covariates were included (Fig. 1). Parameter estimates of models including only postal codes with low SES were lower and not significant ($\beta_{trees}$ = 0.04, SE = 0.03, $p$ = 0.22), compared with all data ($\beta_{trees}$ = 0.06, SE = 0.03, $p$ = 0.04, Fig. 1). Parameter estimates of models including only postal codes with high SES were higher compared with all data and data from postal codes with low SES ($\beta_{trees}$ = 0.08, SE = 0.04, $p$ = 0.04).

Living in a postal code with tree species richness of 14.24, one standard deviation more than the mean (6.83 tree species), increased the likelihood of reporting good mental health by 5.36% (CI = 2.79–7.86%; Fig. 2). Living in a postal code with a bird Shannon diversity of 1.12 – one standard deviation (0.36) more than the mean (1.51) – increased the likelihood of reporting good mental health by 6.64% (CI = 4.11–9.12%; Fig. 2). Parameter estimates of bird and tree diversity were relatively small compared to most socio-demographic and health behavior variables (Fig. 3). The positive association between bird and tree diversity and self-rated mental health was comparable to higher daily fruit and vegetable consumption (Fig. 3).

After adding socioeconomic and health behavior variables in the model, the association with Shannon bird diversity remained relatively stable ($\beta_{birds}$ = 0.06–0.09), while the association with tree diversity attenuated ($\beta_{trees}$ = 0.06–0.11) and distance to blue space became non-significant ($p$ < 0.01 to $p$ = 0.08). No other variables were significant.

**Stress**

Tree and bird diversity was not significantly associated with self-rated stress, regardless of data stratification by neighborhood SES (all $p$ > 0.05). In models with only species diversity and blue and green space variables, including all data and high marginalization data, modeled bird species richness was positively related to the probability of low self-rated stress, but the association was not statistically significant ($\beta_{birds}$ = 0.02, SE = 0.01, $p$ = 0.07 and $\beta_{birds}$ = 0.03, SE = 0.02, $p$ = 0.07). These associations became increasingly weak as socio-demographic and health behaviors were added (Fig. 1).

The most parsimonious model for probability of low self-rated stress included NDVI within the postal code, Shannon diversity of tree species, and modeled bird species richness (AIC = 49023.8, deviance explained = 0.442%, $n$ = 48963, Supplementary Table S2). In all models and data stratifications, NDVI within the postal code was negatively related to low self-rated stress (i.e., higher greenness was associated with higher stress, $\beta_{NDVI}$ = −0.08, SE = 0.03, $p$ = 0.01).

**Discussion**

We found support for a positive association between metrics of bird and tree species diversity and self-rated mental health in postal codes in Canadian cities. This association persisted when covariates, including socio-demographics and health behaviors, were added to models. For bird species diversity, the positive association with self-rated mental health persisted when data were stratified by low and high marginalization. Our results are in line with prior evidence from the UK where higher bird species richness was associated with the prevalence of good health from a national census[30], from Germany where plant and bird species richness was positively related with a mental health index from household questionnaires[31], and from continental

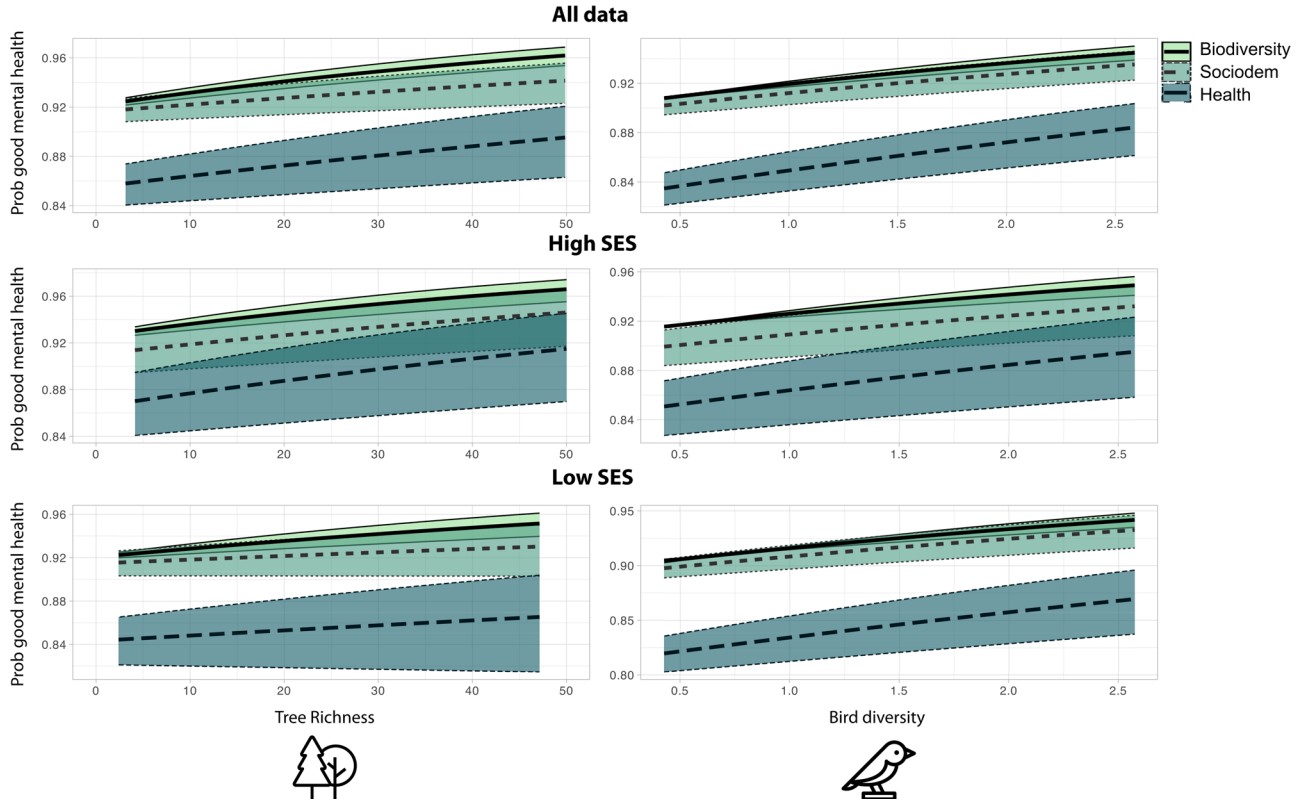

**Fig. 2 | Higher bird species diversity and tree species richness was associated with better self-rated mental health.** The association between the probability of good self-rated mental health in the Canadian Community Health Survey and tree species richness (left panel) and bird Shannon diversity (right panel) predicted from generalized additive models. Shading represents standard error. In "biodiversity" models, only species diversity and greenspace and bluespace variables were included, "sociodem" models added socio-demographic variables, and "health" models added health behavior and socio-demographic variables. All data were included, and data were stratified by neighborhoods with high socio-economic status (low marginalization) and low socio-economic status (high marginalization). Icons from the Noun Project.

trends in Europe, where bird species richness was positively associated with life satisfaction[50]. Our results are also similar to smaller scale experimental studies, in which everyday encounters with birds were associated with time-lasting improvements in mental well-being[51] and listening to recordings with a higher diversity of bird species song were associated with lower depression and listening to any bird song alleviated anxiety and paranoia[52]. The increase in literature showing associations between species diversity and mental health suggests that urban nature-based solutions, including tree planting and avian habitat restoration, represent an opportunity to address both mental health challenges and biodiversity conservation[53].

Three mechanistic pathways by which biodiversity benefits health have been proposed: reducing harm (regulation of air and noise pollution and extreme heat), restoring capacities (restoring psychological and cognitive resources), and building capacities (strengthening of capabilities for meeting everyday demands[27]). Much of the research exploring the associations between biodiversity and health has measured plant and bird diversity[26]. For people living in urban environments, interactions with birds and trees constitute their predominant exposure to wildlife[54]. Birds have conspicuous vocal communication, high densities in urban areas[55], and are important indicators of ecosystem health[56]. Thus birds have great potential for providing restorative benefits and building capacities for humans in cities[57]. Yet, avian populations are facing widespread declines across North America[58] and urbanization is a major driver[59]. Street trees and urban forests are known to provide various ecosystem services for human health and well-being, from air quality to controlling heat (i.e., 'reducing harm' pathway of biodiversity health benefits[60]). Given their rich cultural resonance, trees feature prominently in urban design and urban forests are valued as symbols comfort, peacefulness, beauty, a connection to nature and biodiversity ('restoring and building capacity' through contributing to place identity and

attention restoration[61]). Moreover, the structure of urban forests are known to influence the composition of bird communities[43], although we did not find evidence of a correlation between tree and bird species diversity (Supplementary Fig. S2). While the number of street and park trees is estimated to be increasing in Canadian municipalities, the amount of natural forest cover is decreasing[62], and climate change, diseases, and pests are predicted to increase urban tree mortality[63,64].

We found that the effect size of the association between self-rated mental health and bird and tree species diversity was similar to that of daily servings of fruit and vegetables. Canada's food guide has straightforward recommendations of daily fruit and vegetable intake, including guidance for health professionals and policy makers[65]. No standards or recommendations exist for urban planners to foster bird and tree diversity in urban areas. However, there are a growing number of Park Prescription programs rolling out across Canada, where physicians prescribe patients time spent in nature[66] By working with ecologists, physicians could develop recommendations for patients facing mental health issues to spend more time in areas with higher bird and tree diversity.

We found little evidence that distance to or amount of greenness in postal codes was related to self-rated mental health. We posit that this association is weak because our models include both bird and tree diversity in addition to greenspace and bluespace variables. In public health research, greenspace is generally treated as a homogenous 'natural' environment measured through NDVI[67]. However, urban greenspace and greenness range widely, from forests, lawns, parks, gardens, yards, remnant patches of native vegetation, street trees, and vacant lots of invasive vegetation, each with varying ability to support biodiversity and human health[68]. Increasing evidence shows that attributes of natural environments are fundamental to restorative experiences in greenspace[69]. This can include different land cover

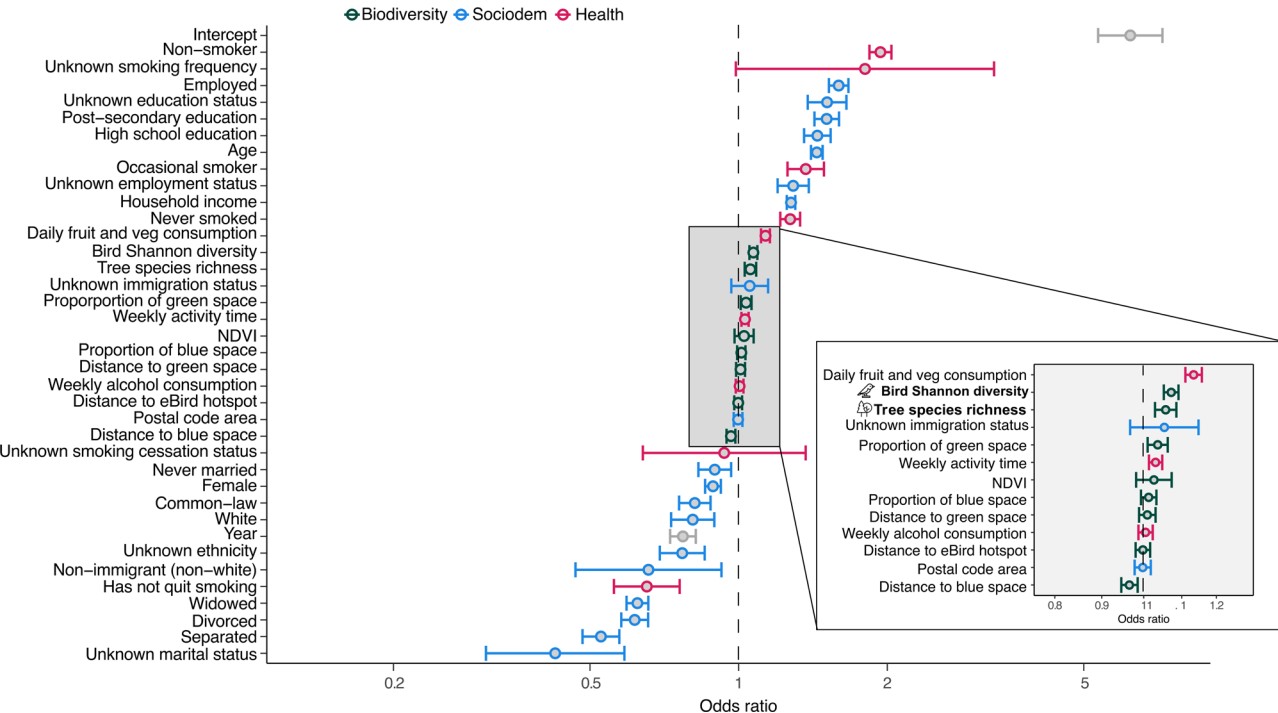

**Fig. 3 | Bird species diversity and tree species richness had a similar independent effect on self-rated mental health as the amount of fruit and vegetables consumed.** Odds ratio of each biodiversity and greenspace and bluespace, socio-demographic, and health behavior variable in a generalized additive model predicting good self-rated mental health of respondents in the Canadian Community Health Survey. Error bars represent standard error. The inset shows a closer view of biodiversity and greenspace and bluespace covariates. Icons from the Noun Project.

types, indicators of management, amenities, infrastructure, signs, safety, and cleanliness[70]. In one Australian study, birds, plants, wildlife, native species and biodiversity were the most important elements that defined people's favorite outdoor places[71]. Given the biophilia hypothesis, it is likely that aspects of nature, such as biodiversity and health of an ecosystem, are better predictors of well-being benefits than greenness[72].

Contrary to our prediction, we found a small increase in the association between bird and tree diversity and self-rated mental health when data were limited to postal codes with lower marginalization, or higher SES (Fig. 1). The diversity of trees was not significantly related to self-rated mental health in postal codes with high marginalization, or lower SES. Some studies have found stronger associations with greenness among participants with lower SES[35,73]. It has been suggested that lower SES groups may benefit more from local greenspace because people in these groups generally have poorer health, offering more opportunity for the health improvements offered by access to nature[34]. However, we found little difference in self-rated mental health and self-rated stress when data were stratified by low and high marginalization. In Europe, one study found that green spaces are more important for the well-being of lower SES residents only in highly urbanized cities[74]. Moreover, a review of place-based interventions found that greening can increase health inequities because those of higher SES are more likely to engage with interventions[75]. In general, findings from other settings may not generalize to Canadian cities, given differences in the distribution of SES and neighborhood influences on health[76]. Socio-demographic factors such as marital, immigration, education, race, and income status were among the strongest predictors of self-rated mental health in our study sample. Thus, adopting a health equity lens and strategies developed alongside communities will be fundamental when planning and managing biodiversity in Canadian cities to maximize equitable health benefits.

We found only weak evidence for an association between bird diversity and self-rated stress and no association between tree diversity and self-rated stress. The lack of association between self-rated stress and species diversity but positive association between self-rated mental health and biodiversity suggests that bird and tree diversity may be affecting mental health through

avenues other than stress, such as positive emotion, happiness, or fascination[77–79]. Thus, although other aspects related to personal experience may be driving stress, biodiverse environments could be related to overall mental health by improving coping, inciting pleasure and enjoyment[80], or providing distance from tasks and thoughts requiring directed attention (i.e., being away[81]), instead requiring effortless, involuntary attention (i.e., fascination, ART[12,82]). For example, research in the UK has found that avian diversity is strongly correlated with emotional response to a space, increasing happiness[83] and that bird watching and feeding benefits mental health by fostering positive emotions[84].

Our analysis was ecological and cross-sectional and relied on self-perceived indicators of mental health, limiting our ability to draw firm causal links between bird and tree diversity and mental health. For example, people with better mental health may choose to live in areas with higher biodiversity. Moreover, birds are indicators of ecosystem health[56]; thus, the association between mental health and species diversity could be driven by other aspects of healthy ecosystems (e.g., reduced exposure to heat) or mediated by health behaviors known to be driven by greenness (e.g., substance abuse[85]). Another caveat is that biodiversity metrics were summarized at the postal code level to correspond with CCHS participants. Although we limited the size of postal codes in our analysis, there may be variation in species diversity within each postal code, differing in accuracy based on the size of the postal code, influencing confidence intervals. We note that models to generate tree species richness were validated against plots in forested areas, meaning estimates may not identify the full set of tree species present in urban areas, including non-native species. This likely resulted in underestimation of tree species diversity in postal codes across CMAs, therefore not biasing our conclusions, but may have reduced the observed impact of tree diversity on mental health. Moreover, forest inventory data come from 2011 only, thus they are likely a noisier representation of species composition than eBird data, which were available across time. Because we only used postal codes with eBird localities with more than nine checklists nearby, our analysis excluded neighborhoods with little or no nearby green space. Thus, in general we acknowledge that

this study population is not completely representative of the full Canadian population. Finally, there is considerable variability in the extent to which individuals have an affinity for nature and ways in which this nature relatedness may affect stress[79]. The effects of birds and trees can vary by person and in some cases may not be beneficial, such as the potential for tree pollen to exacerbate allergy conditions[60]

There is an important need for longitudinal, experimental, and intervention studies that test mechanisms of causation between biodiversity and human health and well-being[86]. However, uncertainty around the mechanisms by which biodiversity influences mental health should not be used as an excuse for inaction[40]. Our national-scale results add to the evidence that shows encounters with wildlife and exposure to biodiversity in urban environments are linked with mental health[26]. It is well understood that access to nature is restricted in cities[87] and urbanization is associated with habitat loss and reduction in native biodiversity[1]. We are at a key juncture - just as we are beginning to appreciate the breadth of human health benefits that are conferred by nature and biodiversity, we are experiencing a biodiversity crisis threatening the vital resources people receive from nature[86]. Conservation of urban greenspace and biodiversity is understood as an increasingly important mechanisms for achieving wider conservation goals (i.e., Target 12 "Significantly increase the area and quality, and connectivity of, access to, and benefits from green and blue spaces in urban and densely populated areas…"[88]). Given the association between tree and bird diversity and self-rated mental health at the urban neighborhood level in Canada, holistic nature-based interventions that bolster biodiversity can be seen as a key tool for public health planning and policy in Canadian municipalities.

## Methods
All analyses were performed in R statistical software version 4.2.1[89].

### Study design and study population
Survey participants are aged 12 years and older, excluding full-time members of the Canadian Armed Forces, residents of Indigenous reserves, and individuals living in institutions. From 2007 onwards, approximately 65,000 respondents were interviewed each year, in 2 year cycles[90]. Approval to access CCHS data was granted by the Carleton University Research Data Center. All data were de-identified and kept in a secure computer facility on campus and thus did not require university ethics approval for use.

For the purpose of this study, data from the 2007 to 2021 CCHS were used (15 CCHS cycles). Because many of the health behavior variables do not apply to participants under 17 (e.g., marital status, smoking, etc.), our analysis includes adults aged 18 years and older. This resulted in a pooled sample of $n = 371,928$ CCHS survey respondents in CMAs in postal codes <16 km$^2$. In the final data set, mean postal code size was 5.8 km$^2$, median was 5.1 km$^2$ with a standard deviation of 3.9 km$^2$. The CCHS underwent a redesign in 2015; thus we only use variables that were consistent before and after.

### Mental health indicators (dependent variables)
The CCHS includes several questions that assess the psychological condition of respondents, including questions related to mental health and stress[49].

Self-rated mental health: Participants were asked about their perceived mental health: "In general, would you say your mental health is 1) Excellent, 2) Very good, 3) Good, 4) Fair, or 5) Poor". We created a binary variable, where values of 1–3 were considered 'good self-rated mental health'[47]. Research by Statistics Canada found that CCHS respondents with mental morbidity had significantly higher odds of reporting fair or poor self-rated mental health than did those not classified with mental morbidity, suggesting that binary self-rated mental health is a useful metric for general mental health[91].

Self-rated stress: To assess perceived stress, participants were asked "Thinking about the amount of stress in your life, would you say that most of your days are 1) not at all stressful, 2) not very stressful, 3) a bit stressful, 4)

quite a bit stressful, or 5) extremely stressful". We also grouped this into a binary variable where values of 1–3 were considered 'low self-rated stress'.

### Biodiversity data (independent variables)
**Bird species richness and diversity.** To generate estimates of bird species richness and diversity we used eBird, as it is one of the largest and most spatially comprehensive publicly available biodiversity datasets[92]. Through the eBird app or website, volunteers submit 'checklists' with the number of individuals of each species observed, start time, duration, and distance covered. Previous studies have shown that when bird sightings are verified by regional reviewers and data are appropriately filtered, eBird checklists can provide reliable data to understand avian diversity in urban greenspaces[93,94]. We note that community scientists involved in assembling eBird checklists in each location need not live in that location - many eBird users travel to perform checklists. The association between an eBird checklist and a location simply means that a registered eBird user traveled to a particular location to perform that survey.

We downloaded Canadian eBird checklists from 2007 to 2021 (https://ebird.org/data/download). We then filtered the checklist data, using methods from[95,96]. We selected checklists that were[1]: complete (where all birds seen and heard were recorded)[2]; had a travel distance <10 km[3]; between 5 and 240 min duration; and[4] were 'stationary', 'traveling', or 'exhaustive'. We removed species that were recorded on fewer than 5% of checklists at each location to limit the impacts of vagrant birds or erroneous identification on the analysis[95]. eBird users can give their own names or choose from existing localities (e.g., eBird hotspots or locations that people regularly visit for birding; https://ebird.org/ebird/hotspots). To estimate the diversity of bird species in neighborhood greenspaces, we calculated the distance to the nearest eBird locality and the bird species diversity metrics of that locality, for each postal code. We removed postal codes with no localities within 1.06 km from postal code edges - the largest estimate of the perceived size of a community in the U.S., based on[97]. At least nine checklists are required to represent species diversity of bird communities with 90% confidence in urban greenspaces[95]; thus, we limited localities to those with a minimum of nine complete checklists in a year. To account for uneven sampling, we generated estimates of species richness and Shannon diversity using rarefaction (generating estimates corresponding to a sample size of 17 checklists[98]) and nonparametric asymptotic estimation[99] in the *iNEXT* package[100].

Given eBird surveys occur at any time of year and checklists can vary in length, we standardized species diversity estimates to 120 min checklists on June 5 using generalized linear model predictions. Although previous research has found more checklists submitted in areas with higher income and higher proportion of white residents, we found little evidence of a relationship between SES and number of checklists in our dataset (see Supplementary Methods for details).

**Tree species diversity.** We extracted tree species richness and diversity from the publicly available National Forest Inventory (NFI). The NFI employs a combination of ground-survey plots, photo identification plots, and Moderate Resolution Imaging Spectroradiometer (MODIS) remote sensing data in a k-Nearest Neighbor (kNN) modeling approach to produce countrywide maps of tree species volume (at 250 m resolution) estimated for 2011. We calculated mean Shannon diversity and species richness of tree species within each postal code polygon using the *diversity* function in the *vegan* package[101].

**Other metrics of green and blue space.** We examined the effect of overall greenness (NDVI) from Moderate Resolution Imaging Spectroradiometer (MODIS) data. These data are compiled as the mean from May 1st through August 31st (growing season) by the Canadian Urban Environmental Health Research Consortium (CANUE) in 500 m and 1000 m circular buffers around each postal code centroid in Canada[102]. We assigned two-year mean NDVI corresponding to, or closest to, the two years of the CCHS survey cycle. We assigned a value of 0 to NDVI

values of NA, as they likely pertained to areas that could not contain green space (e.g., water). We also used the proximity to and proportion of green space and blue space, from the North American Land Change Monitoring System at 30 m from Landsat imagery.

## Covariates

We explored the effect of other characteristics of survey participants known to affect mental health, collected using the CCHS questionnaire (Supplementary Fig. S1). This included socio-demographic characteristics: age, sex, marital status, employment, total household income (ordinal classes treated as a continuous variable), highest level of education attained (treated as a categorical variable), immigration status (White Canadian-born, non-white Canadian born, white immigrant born outside of Canada (0–9 years in Canada), non-white immigrant born outside of Canada (0–9 years in Canada), white immigrant born outside of Canada (10+ years in Canada), non-white immigrant born outside of Canada (10+ years in Canada), and White vs non-White and health behaviors: total daily fruit and vegetable servings consumed per day, smoking status (type of smoker: every day, occasionally, not at all; binary smoking cessation), alcohol consumption, and amount of leisure physical activity (energy expenditure function calculated using the cchsflow package in R statistical software version 4.1.3, a different version was available in the Research Data Center[103]). We included health behaviors as covariates, as they are known to influence mental health[104–106]. Physical activity could be considered a modifier in the relationship between biodiversity and mental health; however, because we lacked information on whether physical activity occurred outside or inside we included it as a covariate. Missing or 'I don't know' responses were included as separate levels of categorical variables to increase the number of complete cases with which to fit the model.

## Moderator

The Canadian Marginalization Index from the Canadian Urban Environmental Health Research Consortium (CANUE) was available at the postal code level, calculated for 2016, and describes material deprivation, residential instability, dependency, and ethnic concentration quantified in continuous values and categorized in quintiles[107].

To account for evidence suggesting that metrics of SES can affect the relationship between biodiversity and mental health[14,39], we stratified our analysis by neighborhood marginalization using the Canadian Marginalization Index from CANUE. We stratified data instead of using an interaction term to decrease model complexity and for ease of model interpretation. We selected the "instability" marginalization dimension, as these models had the best fit with both tree and bird species diversity (Supplementary Methods). We classified postal codes as being 'high marginalization' if they were in the top two quintiles of instability, and as being 'low marginalization' if they were in the bottom three quintiles of instability.

## Analysis

All code and derived data are available at https://doi.org/10.5281/zenodo.11185555 (data privacy restrictions notwithstanding). We linked individual level mental health and sociodemographic data with biodiversity and bluespace/greenspace variables summarized at the postal code level. Due to computational infeasibility of including finer-scale random effects, CMA was included as a random effect to control for local factors leading to clustering in responses.

We fit two sets of logistic GAMs (binomial models with a logit link) using the following response variables: 1) high/low self-rated mental health and 2) high/low self-rated stress (details in Supplementary Methods and Supplementary Table S3). To control for unexplained spatial variability and nonlinear temporal variability in self-rated mental health, CMA was included as a random effect, as well as a continuous smoother fit to CCHS survey year and postal code area. For each response variable we fit three models: 1) only biodiversity and bluespace/greenspace variables; 2) adding in socio-demographic characteristics (age, sex, marital status, income, education, ethnicity, and immigration status); and 3) adding in socio-demographic

characteristics and health behaviors (fruit and vegetable consumption, smoking, alcohol consumption, and physical activity). We fitted these model sets for all data and stratified by low and high marginalization neighborhoods. All continuous variables were centered and scaled by subtracting the mean and dividing by the standard deviation. To assess collinearity among variables, we computed a Spearman's correlation matrix. For variables with a correlation coefficient $R > 0.7$[108] we included one of each covariate in a separate model (where it was the only term) and chose the covariate with the model that resulted in the lowest Akaike's Information Criterion (AIC). We found positive correlations among the following sets of biodiversity and greenspace/bluespace covariates: among different ways of estimating bird species diversity (Chao estimated richness, Chao estimated Shannon diversity, modeled richness, modeled Shannon diversity), tree species richness and tree Shannon diversity, NDVI in different sized buffers (Supplementary Fig. S2). We found no correlation among biodiversity and greenspace/bluespace covariates, health behavior, and sociodemographic data.

Different amounts of missing data meant that each of these models contained a different sample size of complete cases. In the model set with socio-demographic characteristics and health behaviors we found evidence of complete separation (response variable separates the categorical predictor variables perfectly) due to large amounts of missing CCHS data. Thus, we imputed missing values using multiple imputation by chained equations via the *mice* R package following the approach outlined in ref. 109. We compared results from this model to those where missing values were imputed to the median and found little difference in the results (Supplementary Tables S4 and S5).

## Reporting summary

Further information on research design is available in the Nature Portfolio Reporting Summary linked to this article.

## Data availability

All secondary data, code, and materials used in the analyses are available here https://doi.org/10.5281/zenodo.11185555. Note that we are unable to share raw or manipulated health data due to privacy.

## Code availability

All code used in the analyses are available here https://doi.org/10.5281/zenodo.11185555.

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

## Acknowledgements
This research was conducted at the Carleton University Data Center, a part of the Canadian Research Data Center Network (CRDCN). This service is provided through the support of Carleton University, the Canadian Foundation for Innovation, the Canadian Institutes of Health Research, the Social Science and Humanity Research Council, and Statistics Canada. In particular we thank Xuefeng Hu for his guidance through the CRDCN process. The content and views expressed in this article are those of the authors and do not necessarily reflect those of the Government of Canada. We acknowledge generous funding from: National Institutes of Health (National Cancer Institute grant R01CA239197 - A.L.P., R.T.B.) and Natural Sciences and Engineering Research Council of Canada (RGPIN 06147 – J.R.B. and RGPIN 04888 - R.T.B.).

## Author contributions
Conceptualization: R.T.B., P.J.V., E.L. Methodology: E.J.H., R.T.B., J.R.B., S.A.P., T.H., P.J.V., E.L., A.L.P. Visualization: C.R., E.J.H., R.T.B. Supervision: E.J.H., R.T.B., J.R.B. Writing—original draft: E.J.H., R.T.B. Writing—review & editing: E.J.H., R.T.B., J.R.B., S.A.P., A.L.P., P.J.V., E.L., T.H., C.R.

## Competing interests
The authors declare no competing interests.
