## [Peer Review File · Communications Earth & Environment]

8th Feb 24

Dear Dr Buxton,

Your manuscript titled "Mental health is positively associated with biodiversity in Canadian cities" has now been seen by 3 reviewers, and we include their comments at the end of this message. They find your work of interest, but some important points are raised. We are interested in the possibility of publishing your study in *Communications Earth & Environment*, but would like to consider your responses to these concerns and assess a revised manuscript before we make a final decision on publication.

We therefore invite you to revise and resubmit your manuscript, along with a point-by-point response that takes into account the points raised. Please highlight all changes in the manuscript text file. In particular, for publication in *Communications Earth & Environment*, we request that you:

***Provide comprehensive method details, including an explanation of what units are used, scale of postal code areas with descriptive statistics, justify your choice of binary variables, and consider adding a sensitivity analysis.

***Provide an in-depth discussion of study limitations, theory, and evidence on the effect of tree and bird biodiversity on mental health.

Please use the following link to submit your revised manuscript, point-by-point response to the referees' comments (which should be in a separate document to any cover letter), a tracked-changes version of the manuscript (as a PDF file) and the completed checklist:

[link redacted]

We hope to receive your revised paper within six weeks; please let us know if you aren't able to submit it within this time so that we can discuss how best to proceed. If we don't hear from you, and the revision process takes significantly longer, we may close your file. In this event, we will still be happy to reconsider your paper at a later date, as long as nothing similar has been accepted for publication at *Communications Earth & Environment* or published elsewhere in the meantime.

Please do not hesitate to contact us if you have any questions or would like to discuss these revisions further. We look forward to seeing the revised manuscript and thank you for the opportunity to review your work.

Best regards,

Martina Grecequet, PhD
Associate Editor,
Communications Earth & Environment
@CommsEarth

EDITORIAL POLICIES AND FORMATTING

Editorial Policy: Policy requirements (Download the link to your computer as a PDF.)

Furthermore, please align your manuscript with our format requirements, which are summarized on the following checklist:

Communications Earth & Environment formatting checklist

and also in our style and formatting guide Communications Earth & Environment formatting guide .

*** DATA: Communications Earth & Environment endorses the principles of the Enabling FAIR data project (<http://www.copdess.org/enabling-fair-data-project/>). We ask authors to make the data that support their conclusions available in permanent, publically accessible data repositories. (Please contact the editor if you are unable to make your data available).

All Communications Earth & Environment manuscripts must include a section titled "Data Availability" at the end of the Methods section or main text (if no Methods). More information on this policy, is available at <http://www.nature.com/authors/policies/data/data-availability-statements-data-citations.pdf>.

If a community resource is unavailable, data can be submitted to generalist repositories such as figshare or Dryad Digital Repository. Please provide a unique identifier for the data (for example a DOI or a permanent URL) in the data availability statement, if possible. If the repository does not provide identifiers, we encourage authors to supply the search terms that will return the data. For data that have been obtained from publically available sources, please provide a URL and the specific data product name in the data availability statement. Data with a DOI should be further cited in the methods reference section.

REVIEWER COMMENTS:

Reviewer #1 (Remarks to the Author):

Thank you very much for the opportunity to review this paper. It is an interesting investigation of the association between biodiversity and mental health and stress in Canadian cities. I enjoyed reading it and think it is an important contribution to the literature; the findings are interesting and relevant and could inform - together with the wider literature - policy and planning. My main comments would be to introduce biodiversity and common measures of biodiversity in the introduction, to reconsider some of the adjustment variables (e.g., not to adjust for potential mediators), to be a bit clearer about the unit of analysis (i.e., if it is the individual or the postal code), relatedly, think about whether other levels need to be added to the models, and to justify re-coding continuous variables into binary variables (and potentially provide sensitivity analyses for continuous variables). Please find more detailed comments on each section below.

Abstract

- Could this sentence be made clearer/split in two: "Living in a postal code with bird Shannon diversity one standard deviation more than the mean increased likelihood of reporting good mental health by 6.64% and postal codes 22 with tree species richness one standard deviation more than the mean increased the chance of 23 reporting good mental health by 5.36%." What exactly are these percentages? Are these based on odds ratios?

Introduction

- Lines 36 to 40: Could the statistics be explained/clarified? More than 50% of the population in middle- and high-income countries will suffer from at least one mental health disorder, but when exactly (e.g., some time in their life)? And how does that link to the circa 20% of Canadians? Also, what is 8.9 million as a percentage? I suspect more than 20%?

- Lines 41 to 43: Could the statement about the importance of neighbourhood characteristic/geographic inequalities be made a bit more specific? This is a very interesting point, but I am not sure what exactly it means - e.g., is there a percentage or reference or comparison that can be provided?

- Lines 44 to 52: If I remember correctly, the stress recovery theory is more directly related to the biophilia hypothesis than the attention restoration theory. May be worth to double-check this.

- Lines 54 to 56: Are differences between SES groups relevant for this paper? I see it becomes relevant later in the paper, but then I would say a bit more about this in the introduction (i.e., add a paragraph).

- Could the authors add a paragraph on the definition and common measurements of biodiversity? The authors could also add why biodiversity in particular may be relevant for mental health (compared to other measures of nature). E.g., does it increase restorativeness and/or may it be associated with cleaner air?

Results

- Could the authors add sensitivity analyses that are less restrictive and not exclude postal code areas that are especially large? The > 75th percentile criterion makes sense but does seem a bit arbitrary, so seeing results for analyses including all postal areas would be helpful.

- Lines 110 to 114: Why did the authors create binary variables? If they have continuous data, this may be preferable to increase statistical power. Also, I am not sure what the authors mean by probability here?
- Lines 115 to 123: CMA postal codes with sufficient data are the ones included in the analysis, correct? Is this section essentially a bias analysis? Maybe the authors could add a sub-heading for this, so the reader understands better why this paragraph is presented.
- Fig S1: I am not sure I fully understand or agree with the DAG. For example, I would draw an arrow from SES variables to exposure variables, as SES may cause where an individual lives and therefore levels of nature etc. I am also not sure exactly how nature causes marginalization. I would think this again is associated with SES. I see that the authors say marginalization is a modifier (not mediator) but then it should not be presented on the causal pathway between nature and mental health.
- Lines 132 to 136: Are health behaviours really confounders here? They do not cause the exposure (nature). They may, however, be caused by nature, so may lie on the causal pathway between exposure and outcome. Adjusting for these may cause bias (as adjusting for mediators means you are blocking the association between exposure and outcome, thereby decreasing the size of the coefficients).
- Lines 137 to 141: I am not sure what the authors mean by most parsimonious? Do they mean this model had the best fit (considering fit and parsimony)? Compared to what other models? Also, does this mean the authors included multiple exposures in the same model? This can be okay if the authors wanted to mutually adjust for exposures. Can the authors justify this choice?
- Did the authors run interactions between exposures and SES before stratification? I think that may be a good idea. (Also see my later comment on Methods.)
- Lines 157 to 159: I am not sure what this statistic is compared to the statistics presented in the sentences before. This is also tree species richness, correct? Also, I would advise using language that does not imply causality (i.e., association, not effect) - throughout the manuscript.
- Lines 164 to 167: I would advise not to interpret estimates for adjustment variables and also not compare estimates of exposures to these. (Also see my later comments on figures/tables.)
- Associations with stress: I would report that the authors did not find evidence for associations but refrain from using terms like "near-statistically significant". If the authors use significance as a decision tool, they can follow the decision rule $< .05$ (or whatever cut off they decided to use) but then anything larger than $.05$ is simply not significant. Others may advise not to refer to significance at all, but I think this is the authors' choice.
- Figure 2: I see that the size of coefficients decreases with adjustment. For the adjustment for SES variables, this makes sense and is important (as it shows potential confounding by SES). However, for the health variables, I would think these could also be mediators in the association, so adjustment for them may mean bias (i.e., decrease of size of estimates due to overadjustment).
- Figure 3: Adjustment variables should not be interpreted (as they are just included for adjustment, not as exposures, and their estimates may not be true, as models were not modelled and adjusted to investigate these variables as exposures).
- Tables S3 and S4: For the same reason, estimates for adjustment variables should not be presented in tables.

Discussion

- Lines 210 to 217: I thought this was a bit difficult to follow because the authors discussed the importance of bird species diversity but then moved to discussing the importance of trees here. Maybe they wanted to say that trees offer important habitat for birds - then they could make that clearer.

- Lines 218 to 226: The authors make a very good point here. The only thing I would not do (and I mentioned this earlier) is compare the size of estimates for exposure and adjustment variables. I think the authors can make the same point without this.
- Lines 274 to 275: Very important point. Do I understand correctly that data were linked at postal code level, not individual level? If it was at individual level, the authors could adjust for baseline mental health, as they seem to have longitudinal data. (Also see my comments on Methods.)

Methods

- Lines 331 to 339: Why did the authors create binary variables? I would add a sensitivity analysis using continuous outcomes.
- Lines 406 to 408: Did the authors consider multiple imputation (rather than a complete case analysis)? Missing data and 'I don't know' responses could be due to many reasons, so I am not sure these categories are meaningful? What was the pattern of missing data (i.e., how many people had missing data on these variables)? I've read later that the authors did use multiple imputation. Then they can impute data on these variables too (rather than including missing data as a category).
- Lines 419 to 421: Why did the authors recode the marginalization variable into a binary variable? I see this may be useful for stratification, but using a continuous variable and running interactions first may be good to avoid loss of variation/information.
- I am unsure whether models included aggregated data for postal codes (and that is why the CMA was used as Level 2) or whether the models included data at the individual level. If the latter, the authors may want to add a level for the postal code too (i.e., individual, postal code, CMA) - as CMAs are larger than postal codes, correct? Also, if the unit of observation was the individual, not the postal code, the authors may have had repeated measures due to using data of multiple years. If this is true, the authors should add a level for the individual. Also, this would mean that the authors could run a longitudinal model where exposure and confounders are measured at baseline, also adjusted for baseline mental health, and the outcome at follow-up. This would improve the study. In any case, the authors should be very clear about the unit of analysis, as I am not sure which one it is. If it is the individual, adjusting for clustering of observations in individuals would be important.

Reviewer #2 (Remarks to the Author):

General

This study links data from the cross-sectional Canadian Community Health Survey to small area indicators of biodiversity (bird and tree species richness) around participant's homes in 36 Canadian cities. It assesses whether these indicators are associated with self-rated mental health outcomes for adults, and overall finds positive associations. This is a novel addition to the field making use of some large, robust data resources. While the cross-sectional nature of the data limits causal inference, it does give some useful indication that more biodiverse environments might promote better mental health in this context.

Recommendations

1. Intro Line 44-57: This review could do with a bit more critical discussion of the literature I think. I agree that we now have a pretty convincing amount of evidence on nature and mental health, but it's not without its limitations (see pretty much any systematic review on the topic). Also Biophilia

and other theories are open to challenge (e.g. <https://doi.org/10.3197/096327111X12997574391724> or <https://doi.org/10.1080/10937404.2018.1505571>).

2. Methods Line 313: overall the description of methods is good and usefully detailed, and analyses include appropriate robustness checks. However one key issue that pertains to exposure assessment, which is a critical aspect of the robustness of the study, is the geographical scale of the analysis. It's not entirely clear what the scale of a typical postal code area is, although respondents in postal code areas larger than 16 km² were excluded. It would be useful to be clear here on the average size of a postal code area, the minimum, maximum, standard deviation and so on. Following on from that what are the limitations and potential errors of the exposure assessment for environmental metrics? Given that the bird metric is based on the closest eBird locality with a limit of 1.06 km, I'm guessing that the bird metric is more accurate for smaller versus larger postal code areas and therefore in inner city areas versus more peripheral areas? Similarly the NDVI metric based on a 500 metre centroid buffer is likely to be less representative for larger postal code areas? These kinds of issues are discussed a little in limitations, but this could be expanded upon.

3. Results general: Even though the limitations are clearly acknowledged elsewhere, in some places causal language is used i.e. "effect" – relationships should probably be described as associations, as per the paper title.

4. Results Line 103+: it would be useful here to have a really clear statement of the starting n, the exclusions, and the ultimate analysis sample size.

5. Results Line 114: I think this should be low self-rated stress?

6. Results Line 170-172: This is a bit confusing as it describes that distance to blue space became non-significant and then the next sentence suggests distance to blue space had a negative effect? Also "Decreased distance to blue space had a negative effect on self-rated mental health" is not really clear on whether this means living closer to blue space is associated with better or worse mental health – just needs rephrasing.

7. Results Line 125+ (and elsewhere): I'd hesitate to call this a DAG, which is I think more typically used for causal inference and treats each variable individually. I think this is probably better termed a conceptual model or similar?

8. Discussion: A key element that is missing for me (also perhaps warranting a mention in the introduction) is a substantive discussion of hypothesised mechanisms. There are a couple of hints in the discussion e.g. evidence on avian diversity and emotional responses, but I'd like to see a more substantive section of the discussion dedicated to theory and evidence on why we might expect tree and bird biodiversity to promote mental health above and beyond more generic 'greenspace'.

9. Fig 3: Is a useful summary of the main results. I did wonder whether it would be easier to read if variables were grouped together though (i.e. all smoking indicators together, biodiversity/green space all together etc.), I'm not sure having them ordered by OR magnitude is particularly useful? I'm happy to be convinced otherwise though.

Reviewer #3 (Remarks to the Author):

In this manuscript, the authors investigate the association between biodiversity in Canadian cities and indicators of mental health.

The introduction is well written. It briefly discusses the main pathways and theories that link biodiversity to health benefits. The objectives are clear. In addition to the main objective of exploring the relationship between biodiversity indicators (trees and birds) and mental health, the authors also include appropriate stratified analyses (by SES). The authors have used appropriate statistical techniques and have properly adjusted models for SES and behavioral background variables, which are known to have an impact on the relationship between green space and health. The authors found strong evidence for a protective effect of bird diversity on mental health. The evidence for tree diversity was weaker, especially in low SES strata or in models adjusted for all background variables. Nevertheless, the effect size of trees was higher than the effect size of birds. As elsewhere reported, the effect sizes of background variables were larger than those of bird and tree diversity. I particularly liked that the authors could compare the effect size of bird and tree diversity to the effect of daily fruit and vegetable consumption - this is a message that can be picked up easily by media and press, increasing the impact of the manuscript. The risk associated with decreasing distance to blue space is in contrast to earlier findings. Also the association between higher greenness and higher stress is not in line with most literature, unless higher greenness could be seen as an indicator for lower SES in this case.

The discussion is well referenced, follows a logical structure and is well written, offering clear management implications (eg tree planting and avian habitat restoration, represent an opportunity to address both mental health challenges and biodiversity conservation (again, something that can be picked up by media). The authors acknowledge the limitations of the ecological design and the self-reported nature of the health indicators, as well as the potential bias caused by individual variation in nature-relatedness of inhabitants of the studied areas.

COMMSENV-23-1923-T

Title: Mental health is positively associated with biodiversity in Canadian cities

Review

Thank you very much for the opportunity to review this paper. It is an interesting investigation of the association between biodiversity and mental health and stress in Canadian cities. I enjoyed reading it and think it is an important contribution to the literature; the findings are interesting and relevant and could inform - together with the wider literature - policy and planning. My main comments would be to introduce biodiversity and common measures of biodiversity in the introduction, to reconsider some of the adjustment variables (e.g., not to adjust for potential mediators), to be a bit clearer about the unit of analysis (i.e., if it is the individual or the postal code), relatedly, think about whether other levels need to be added to the models, and to justify re-coding continuous variables into binary variables (and potentially provide sensitivity analyses for continuous variables). Please find more detailed comments on each section below.

Abstract

- Could this sentence be made clearer/split in two: "Living in a postal code with bird Shannon diversity one standard deviation more 21 than the mean increased likelihood of reporting good mental health by 6.64% and postal codes 22 with tree species richness one standard deviation more than the mean increased the chance of 23 reporting good mental health by 5.36%." What exactly are these percentages? Are these based on odds ratios?

Introduction

- Lines 36 to 40: Could the statistics be explained/clarified? More than 50% of the population in middle- and high-income countries will suffer from at least one mental health disorder, but when exactly (e.g., some time in their life)? And how does that link to the circa 20% of Canadians? Also, what is 8.9 million as a percentage? I suspect more than 20%?
- Lines 41 to 43: Could the statement about the importance of neighbourhood characteristic/geographic inequalities be made a bit more specific? This is a very interesting point, but I am not sure what exactly it means - e.g., is there a percentage or reference or comparison that can be provided?

- Lines 44 to 52: If I remember correctly, the stress recovery theory is more directly related to the biophilia hypothesis than the attention restoration theory. May be worth to double-check this.
- Lines 54 to 56: Are differences between SES groups relevant for this paper? I see it becomes relevant later in the paper, but then I would say a bit more about this in the introduction (i.e., add a paragraph).
- Could the authors add a paragraph on the definition and common measurements of biodiversity? The authors could also add why biodiversity in particular may be relevant for mental health (compared to other measures of nature). E.g., does it increase restorativeness and/or may it be associated with cleaner air?

Results

- Could the authors add sensitivity analyses that are less restrictive and not exclude postal code areas that are especially large? The > 75th percentile criterion makes sense but does seem a bit arbitrary, so seeing results for analyses including all postal areas would be helpful.
- Lines 110 to 114: Why did the authors create binary variables? If they have continuous data, this may be preferable to increase statistical power. Also, I am not sure what the authors mean by probability here?
- Lines 115 to 123: CMA postal codes with sufficient data are the ones included in the analysis, correct? Is this section essentially a bias analysis? Maybe the authors could add a sub-heading for this, so the reader understands better why this paragraph is presented.
- Fig S1: I am not sure I fully understand or agree with the DAG. For example, I would draw an arrow from SES variables to exposure variables, as SES may cause where an individual lives and therefore levels of nature etc. I am also not sure exactly how nature causes marginalization. I would think this again is associated with SES. I see that the authors say marginalization is a modifier (not mediator) but then it should not be presented on the causal pathway between nature and mental health.
- Lines 132 to 136: Are health behaviours really confounders here? They do not cause the exposure (nature). They may, however, be caused by nature, so may lie on the causal pathway between exposure and outcome. Adjusting for these may cause bias (as adjusting for mediators means you are blocking the association between exposure and outcome, thereby decreasing the size of the coefficients).
- Lines 137 to 141: I am not sure what the authors mean by most parsimonious? Do they mean this model had the best fit (considering fit and parsimony)? Compared to

what other models? Also, does this mean the authors included multiple exposures in the same model? This can be okay if the authors wanted to mutually adjust for exposures. Can the authors justify this choice?

- Did the authors run interactions between exposures and SES before stratification? I think that may be a good idea. (Also see my later comment on Methods.)
- Lines 157 to 159: I am not sure what this statistic is compared to the statistics presented in the sentences before. This is also tree species richness, correct? Also, I would advise using language that does not imply causality (i.e., association, not effect) - throughout the manuscript.
- Lines 164 to 167: I would advise not to interpret estimates for adjustment variables and also not compare estimates of exposures to these. (Also see my later comments on figures/tables.)
- Associations with stress: I would report that the authors did not find evidence for associations but refrain from using terms like “near-statistically significant”. If the authors use significance as a decision tool, they can follow the decision rule $< .05$ (or whatever cut off they decided to use) but then anything larger than $.05$ is simply not significant. Others may advise not to refer to significance at all, but I think this is the authors’ choice.
- Figure 2: I see that the size of coefficients decreases with adjustment. For the adjustment for SES variables, this makes sense and is important (as it shows potential confounding by SES). However, for the health variables, I would think these could also be mediators in the association, so adjustment for them may mean bias (i.e., decrease of size of estimates due to overadjustment).
- Figure 3: Adjustment variables should not be interpreted (as they are just included for adjustment, not as exposures, and their estimates may not be true, as models were not modelled and adjusted to investigate these variables as exposures).
- Tables S3 and S4: For the same reason, estimates for adjustment variables should not be presented in tables.

Discussion

- Lines 210 to 217: I thought this was a bit difficult to follow because the authors discussed the importance of bird species diversity but then moved to discussing the importance of trees here. Maybe they wanted to say that trees offer important habitat for birds - then they could make that clearer.

- Lines 218 to 226: The authors make a very good point here. The only thing I would not do (and I mentioned this earlier) is compare the size of estimates for exposure and adjustment variables. I think the authors can make the same point without this.
- Lines 274 to 275: Very important point. Do I understand correctly that data were linked at postal code level, not individual level? If it was at individual level, the authors could adjust for baseline mental health, as they seem to have longitudinal data. (Also see my comments on Methods.)

Methods

- Lines 331 to 339: Why did the authors create binary variables? I would add a sensitivity analysis using continuous outcomes.
- Lines 406 to 408: Did the authors consider multiple imputation (rather than a complete case analysis)? Missing data and 'I don't know' responses could be due to many reasons, so I am not sure these categories are meaningful? What was the pattern of missing data (i.e., how many people had missing data on these variables)? I've read later that the authors did use multiple imputation. Then they can impute data on these variables too (rather than including missing data as a category).
- Lines 419 to 421: Why did the authors recode the marginalization variable into a binary variable? I see this may be useful for stratification, but using a continuous variable and running interactions first may be good to avoid loss of variation/information.
- I am unsure whether models included aggregated data for postal codes (and that is why the CMA was used as Level 2) or whether the models included data at the individual level. If the latter, the authors may want to add a level for the postal code too (i.e., individual, postal code, CMA) - as CMAs are larger than postal codes, correct? Also, if the unit of observation was the individual, not the postal code, the authors may have had repeated measures due to using data of multiple years. If this is true, the authors should add a level for the individual. Also, this would mean that the authors could run a longitudinal model where exposure and confounders are measured at baseline, also adjusted for baseline mental health, and the outcome at follow-up. This would improve the study. In any case, the authors should be very clear about the unit of analysis, as I am not sure which one it is. If it is the individual, adjusting for clustering of observations in individuals would be important.

We therefore invite you to revise and resubmit your manuscript, along with a point-by-point response that takes into account the points raised. Please highlight all changes in the manuscript text file. In particular, for publication in Communications Earth & Environment, we request that you:

*****Provide comprehensive method details, including an explanation of what units are used, scale of postal code areas with descriptive statistics, justify your choice of binary variables, and consider adding a sensitivity analysis.**

*****Provide an in-depth discussion of study limitations, theory, and evidence on the effect of tree and bird biodiversity on mental health.**

Thank you for these helpful general comments. Please see detailed responses to Reviewer comments below for changes we made to address them.

Reviewer #1 (Remarks to the Author):

Thank you very much for the opportunity to review this paper. It is an interesting investigation of the association between biodiversity and mental health and stress in Canadian cities. I enjoyed reading it and think it is an important contribution to the literature; the findings are interesting and relevant and could inform - together with the wider literature - policy and planning. My main comments would be to introduce biodiversity and common measures of biodiversity in the introduction, to reconsider some of the adjustment variables (e.g., not to adjust for potential mediators), to be a bit clearer about the unit of analysis (i.e., if it is the individual or the postal code), relatedly, think about whether other levels need to be added to the models, and to justify re-coding continuous variables into binary variables (and potentially provide sensitivity analyses for continuous variables). Please find more detailed comments on each section below.

Thank you very much for your positive feedback on our paper, it is much appreciated! Please see below, where we address specific comments.

Abstract

- Could this sentence be made clearer/split in two: “Living in a postal code with bird Shannon diversity one standard deviation more 21 than the mean increased likelihood of reporting good mental health by 6.64% and postal codes 22 with tree species richness one standard deviation more than the mean increased the chance of 23 reporting good mental health by 5.36%.” What exactly are these percentages? Are these based on odds ratios?

We split this into two sentences and clarified. These percentages are calculated based on the fitted odds ratios. We refer to effect sizes relative to the mean and standard deviation of predictors because predictor variables were centred and scaled prior to analysis.

“Living in a postal code with bird diversity one standard deviation higher than the mean increased reporting of good mental health by 6.64%. Postal codes with tree species richness one standard deviation more than the mean increased reporting of good mental health by 5.36%.”

Introduction

- Lines 36 to 40: Could the statistics be explained/clarified? More than 50% of the population in middle- and high-income countries will suffer from at least one mental health disorder, but when exactly (e.g., some time in their life)? And how does that link to the circa 20% of Canadians? Also, what is 8.9 million as a percentage? I suspect more than 20%?

We clarified on line 37:

“More than 50% of the population in middle- and high-income countries will suffer from at least one mental health disorder *at some point in their lives.*”

The 50% of the general population and 20% of Canadians are unrelated – they come from different sources. 8.9 million is 23% of the current Canadian population.

- Lines 41 to 43: Could the statement about the importance of neighbourhood characteristic/geographic inequalities be made a bit more specific? This is a very interesting point, but I am not sure what exactly it means - e.g., is there a percentage or reference or comparison that can be provided?

We edited this statement to reflect the results of Belsky et al. 2019:

“Neighborhood characteristics and geographic inequalities *predominantly* explain mental health outcomes, particularly the characteristics of the urban environment.”

- Lines 44 to 52: If I remember correctly, the stress recovery theory is more directly related to the biophilia hypothesis than the attention restoration theory. May be worth to double-check this.

Thank you for this question and encouraging us to double check.

From our understanding, there are ties to evolution in both ART and SRT. From Stephen Kaplan’s 1995 paper on ART:

“Further, much of what was important to the evolving human-wild animals, danger, caves, blood, to name a few examples-was (and still is) innately fascinating and thus does not require directed attention.”

However, after referring to Terry Hartig’s recent book chapter, neither ART or SRT were related to the Biophilia hypothesis (https://link.springer.com/chapter/10.1007/978-3-030-69020-5_5).

Thus, we have edited the sentence on line 44 to:

“Two key psychological theories *posit the relationship between human health and wellbeing and natural environments*: ‘attention restoration theory’ ...”

We also reordered the text to include the Biophilia Hypothesis in another paragraph, based on the comment below.

- Lines 54 to 56: Are differences between SES groups relevant for this paper? I see it becomes relevant later in the paper, but then I would say a bit more about this in the introduction (i.e., add a paragraph).

We added a paragraph on line 74:

“Associations between nature and health can vary between socio-economic status (SES) groups. Some studies have found the benefits of nature are greater in lower SES neighborhoods (20-22). *Much research has shown that low-income neighborhoods have reduced greenspace availability and biodiversity, termed the ‘luxury effect’ (33). Moreover, residents of lower SES are less likely to use greenspace that does exist (34). However, inequality in biodiversity distribution can be driven instead by urban form, social policy, or collective human preference (35). Studies of the relationship between health and biodiversity often occur in more affluent populations, precluding generalizable relationships between health, SES, and nature (36).*

- Could the authors add a paragraph on the definition and common measurements of biodiversity? The authors could also add why biodiversity in particular may be relevant for mental health (compared to other measures of nature). E.g., does it increase restorativeness and/or may it be associated with cleaner air?

We have rearranged and added some text to speak to biodiversity and its relevance for mental health on line 53:

However, the role of biodiversity and different components of the natural environment in mental health research is unresolved (25). *Biodiversity is defined as the variety of life on Earth, from genes to ecosystems, associated with ecosystem functioning, resilience, and health (26). Biodiversity is commonly measured in terms of species or taxonomic richness and diversity over a geographic area at a particular time (27).* The restorative effects of biodiversity are thought to relate to human evolution, throughout which our species has relied on a variety of species for survival and reproduction (12). Thus, the ‘biophilia hypothesis’ posits that humans have an innate affinity to connect with other species and nature (13). *Higher biodiversity may suggest a safe environment, where a variety of species mean that our needs are being met, leading to psychological restoration and perceived restorativeness (28).*

Results

- Could the authors add sensitivity analyses that are less restrictive and not exclude postal

code areas that are especially large? The > 75th percentile criterion makes sense but does seem a bit arbitrary, so seeing results for analyses including all postal areas would be helpful.

We did not include an analysis with all postal codes because we felt this small number of very large postal codes would have caused higher error rates. Moreover, the majority of these large postal codes were in more rural areas, outside city centres, which was beyond the scope of our study.

- Lines 110 to 114: Why did the authors create binary variables? If they have continuous data, this may be preferable to increase statistical power. Also, I am not sure what the authors mean by probability here?

We created binary variables based on research by Statistics Canada that found participants who rated mental health as “fair” or “poor” (considered in this study as poor self-rated mental health) had significantly higher odds of mental morbidity (Line 354).

We edited line 125:

“The *proportion of participants who reported* good mental health was 0.92 (standard deviation, SD=0.27) and the *proportion of participants who reported* low stress was 0.80 (SD=0.40, Table S1).”

- Lines 115 to 123: CMA postal codes with sufficient data are the ones included in the analysis, correct? Is this section essentially a bias analysis? Maybe the authors could add a sub-heading for this, so the reader understands better why this paragraph is presented.

We have added a sub-heading called “sampling bias” here and a sentence on line 129:

“*This suggests an overrepresentation of wealthier, more active and older people who consume fewer alcoholic drinks within our sample.*”

- Fig S1: I am not sure I fully understand or agree with the DAG. For example, I would draw an arrow from SES variables to exposure variables, as SES may cause where an individual lives and therefore levels of nature etc. I am also not sure exactly how nature causes marginalization. I would think this again is associated with SES. I see that the authors say marginalization is a modifier (not mediator) but then it should not be presented on the causal pathway between nature and mental health.

We tested for the effect of SES on species diversity and found little evidence of an effect of marginalization on bird diversity (see supplementary material):

“Using a generalized linear model with a Poisson error structure we found little evidence of a relationship between metrics of marginalization and number of checklists ($R^2 < 0.02$).”

Based on our reading of Morrow et al 2022 and understanding of how SES can alter the relationship between biodiversity and mental health, we posit that marginalization is a moderator. We have adjusted the arrow to sit above the arrow between exposure and mental health. As per reviewer 2 we have renamed the DAG a ‘conceptual model’, as it is meant to guide our analysis.

- Lines 132 to 136: Are health behaviours really confounders here? They do not cause the exposure (nature). They may, however, be caused by nature, so may lie on the causal pathway between exposure and outcome. Adjusting for these may cause bias (as adjusting for mediators means you are blocking the association between exposure and outcome, thereby decreasing the size of the coefficients).

Thank you for this question. We reviewed this paper: Morrow et al 2022 and posit that health behaviours (fruit and vegetable consumption, smoking, alcohol consumption, and physical activity) are covariates, rather than confounders, as they affect the outcome (mental health). We can find no evidence that fruit and vegetable consumption, smoking, and alcohol consumption are caused by nature. If the reviewer can provide a reference, we would be happy to include it.

Outdoor physical activity may be a mediator or moderator in the relationship between biodiversity and mental health, but as we describe below, we were unsure if the physical activity reported in the CCHS occurs outside or inside, thus we left this as a covariate. To clarify, we have changed ‘confounder’ to ‘covariate’ throughout and include a sentence in the methods on line 428:

“We included health behaviors as covariates, as they are known to influence mental health (101-103). Physical activity could be considered a moderator in the relationship between biodiversity and mental health; however, because we lacked information on whether physical activity occurred outside or inside we included it as a covariate.”

- Lines 137 to 141: I am not sure what the authors mean by most parsimonious? Do they mean this model had the best fit (considering fit and parsimony)? Compared to what other models? Also, does this mean the authors included multiple exposures in the same model? This can be okay if the authors wanted to mutually adjust for exposures. Can the authors justify this choice?

The most parsimonious model based on the lowest AIC is typically one that neither underfits nor overfits (penalized for degrees of freedom of the explanatory variables). We used the model with the lowest AIC to compare amongst greenness and species diversity variables (Table S2). To clarify we state on line 156:

“lowest AIC=26185.90, deviance explained=1.52%, n=47623, compared to models including other biodiversity and greenness variables Table S2”

- Did the authors run interactions between exposures and SES before stratification? I think that may be a good idea. (Also see my later comment on Methods.)

We did not run interactions between exposures and SES. We opted for stratification because we wanted to keep models relatively simple and easy to interpret. We expected a difference between low and high SES in fitted parameter estimates with many of the model terms (i.e., other covariates) beyond the exposure and wanted to control for this without adding interactions for several terms.

- Lines 157 to 159: I am not sure what this statistic is compared to the statistics presented in the sentences before. This is also tree species richness, correct? Also, I would advise using language that does not imply causality (i.e., association, not effect) - throughout the manuscript.

We deleted this sentence, which was out of place and only applied to one model. We have also gone through the manuscript to eliminate causal wording.

- Lines 164 to 167: I would advise not to interpret estimates for adjustment variables and also not compare estimates of exposures to these. (Also see my later comments on figures/tables.)

We believe this is useful, even if not central to our question, to put effect sizes in context. We do so to provide more conservatism to our narrative (i.e., the effect size of species diversity of birds is small compared to some of the key drivers of mental health, i.e., employment status, education, marital status). We have deleted supplementary tables as per the reviewers suggestion.

- Associations with stress: I would report that the authors did not find evidence for associations but refrain from using terms like “near-statistically significant”. If the authors use significance as a decision tool, they can follow the decision rule $< .05$ (or whatever cut off they decided to use) but then anything larger than $.05$ is simply not significant. Others may advise not to refer to significance at all, but I think this is the authors’ choice.

We changed this to (line 188):

“modelled bird species richness was positively related to the probability of low self-rated stress, but the association was not statistically significant”

- Figure 2: I see that the size of coefficients decreases with adjustment. For the adjustment for SES variables, this makes sense and is important (as it shows potential confounding by SES). However, for the health variables, I would think these could also be mediators in the association, so adjustment for them may mean bias (i.e., decrease of size of estimates due to overadjustment).

The purpose of including health behaviours in models was to explore and remove the variation caused by these covariates. Physical activity may be a moderator or mediator; however, we are unsure of how smoking and alcohol would mediate the relationship between species diversity

and mental health. We chose to include physical activity as a covariate in this case because we did not have any information about whether activity was indoors or outside. Moreover, there is little evidence of how physical activity acts as a mediator of biodiversity-human health relationships (Marselle et al 2021 <https://doi.org/10.1016/j.envint.2021.106420>).

- Figure 3: Adjustment variables should not be interpreted (as they are just included for adjustment, not as exposures, and their estimates may not be true, as models were not modelled and adjusted to investigate these variables as exposures).

We emphasize that we use these covariates to remove their variation from the models and to put the biodiversity/green space exposure results in context of other variables that are known to affect mental health.

We have deleted tables S3 and S4 and extended results in the supplementary material, to make it clear that we are not interpreting these adjustment variables alone. Instead, we explore the parameter estimates of species diversity of birds in relation to other covariates (Figure 3).

- Tables S3 and S4: For the same reason, estimates for adjustment variables should not be presented in tables.

We have deleted these tables.

Discussion

- Lines 210 to 217: I thought this was a bit difficult to follow because the authors discussed the importance of bird species diversity but then moved to discussing the importance of trees here. Maybe they wanted to say that trees offer important habitat for birds - then they could make that clearer.

Here we discuss the benefits of trees in urban settings, given we found support for a positive association between metrics of bird AND tree species diversity and self-rated mental health. We added the following sentence to incorporate this feedback (line 232):

“Moreover, the structure of urban forests are known to influence the composition of bird communities (41), although we did not find evidence of a correlation between tree and bird species diversity (Fig. S2).”

- Lines 218 to 226: The authors make a very good point here. The only thing I would not do (and I mentioned this earlier) is compare the size of estimates for exposure and adjustment variables. I think the authors can make the same point without this.

We are unsure of how to make this point without comparing the parameter estimates between exposure and covariates.

- Lines 274 to 275: Very important point. Do I understand correctly that data were linked at postal code level, not individual level? If it was at individual level, the authors could adjust for baseline mental health, as they seem to have longitudinal data. (Also see my comments on Methods.)

Individual level data were linked to biodiversity data at the postal code level (see our response in the methods).

Methods

- Lines 331 to 339: Why did the authors create binary variables? I would add a sensitivity analysis using continuous outcomes.

We created a binary variable because in practice, measures of mental health are commonly defined and reported in this way (e.g., for the purposes of identifying patients who are clinically depressed or not) – see Crouse et al 2021. *Environmental Research* **192**: 110267 and the Canadian Positive Mental Health Indicator Framework (<https://health-infobase.canada.ca/positive-mental-health/Index>). Moreover, research by Statistics Canada found that CCHS respondents with mental morbidity had significantly higher odds of reporting fair or poor self-rated mental health than did those not classified with mental morbidity. To clarify, we added on line 354:

“Research by Statistics Canada found that CCHS respondents with mental morbidity had significantly higher odds of reporting fair or poor self-rated mental health than did those not classified with mental morbidity, suggesting that *binary* self-rated mental health is a *useful metric* for general mental health.”

- Lines 406 to 408: Did the authors consider multiple imputation (rather than a complete case analysis)? Missing data and ‘I don’t know’ responses could be due to many reasons, so I am not sure these categories are meaningful? What was the pattern of missing data (i.e., how many people had missing data on these variables)? I’ve read later that the authors did use multiple imputation. Then they can impute data on these variables too (rather than including missing data as a category).

We performed multiple imputation for continuous variables. For categorical variables we left missing data as ‘I don’t know’. Given we found little difference between interpolation methods, we do not expect that this will have a large effect on adjustment variables.

- Lines 419 to 421: Why did the authors recode the marginalization variable into a binary variable? I see this may be useful for stratification, but using a continuous variable and running interactions first may be good to avoid loss of variation/information.

We opted for stratification because we wanted to keep models relatively simple and easy to interpret. We expected different relationships with multiple variables between strata that could not be captured by a single pairwise interaction with the exposure.

- I am unsure whether models included aggregated data for postal codes (and that is why the CMA was used as Level 2) or whether the models included data at the individual level. If the latter, the authors may want to add a level for the postal code too (i.e., individual, postal code, CMA) - as CMAs are larger than postal codes, correct? Also, if the unit of observation was the individual, not the postal code, the authors may have had repeated measures due to using data of multiple years. If this is true, the authors should add a level for the individual. Also, this would mean that the authors could run a longitudinal model where exposure and confounders are measured at baseline, also adjusted for baseline mental health, and the outcome at follow-up. This would improve the study. In any case, the authors should be very clear about the unit of analysis, as I am not sure which one it is. If it is the individual, adjusting for clustering of observations in individuals would be important.

While we agree that a longitudinal model would be stronger, this is a repeated cross-sectional study. While it is possible that there are repeat respondents captured in the CCHS, this would likely represent a very small proportion and Statistics Canada does not link responses over time.

Models included data for each individual, but biodiversity and bluespace/greenspace variables were aggregated at the postal code level. We attempted to fit models with postal code as a random effect smoother, but they became infeasibly complex to fit (i.e., beyond the capabilities of the servers we used for analysis), so we include CMA as a random variable to account for local factors.

To clarify we added the following on line 449:

“We linked individual level mental health and sociodemographic data with biodiversity and bluespace/greenspace variables summarized at the postal code level. Due to computational infeasibility of including finer-scale random effects, CMA was included as a random effect to control for local factors leading to clustering in responses”

Reviewer #2 (Remarks to the Author):

General

This study links data from the cross-sectional Canadian Community Health Survey to small area indicators of biodiversity (bird and tree species richness) around participant’s homes in 36 Canadian cities. It assesses whether these indicators are associated with self-rated mental health outcomes for adults, and overall finds positive associations. This is a novel addition to the field making use of some large, robust data resources. While the

cross-sectional nature of the data limits causal inference, it does give some useful indication that more biodiverse environments might promote better mental health in this context.

Thank you for these positive comments!

Recommendations

1. Intro Line 44-57: This review could do with a bit more critical discussion of the literature I think. I agree that we now have a pretty convincing amount of evidence on nature and mental health, but it's not without its limitations (see pretty much any systematic review on the topic). Also Biophilia and other theories are open to challenge (e.g. <https://doi.org/10.3197/096327111X12997574391724> or <https://doi.org/10.1080/10937404.2018.1505571>).

Thank you for these helpful references. We agree that there has been a healthy debate about the Biophilia hypothesis. We aimed for a summary of what is known about the relationship between mental health and biodiversity, without getting into the weeds of the debate. We have edited this sentence (and moved it to a new paragraph – line 57):

“The restorative effects of biodiversity thought to relate to human evolution, throughout which our species has relied on a variety of species for survival and reproduction (12). Thus, the ‘Biophilia Hypothesis’ posits that humans have an innate affinity to connect with other species and nature (13, *although for a critique of the Biophilia Hypothesis see 28*).”

2. Methods Line 313: overall the description of methods is good and usefully detailed, and analyses include appropriate robustness checks. However one key issue that pertains to exposure assessment, which is a critical aspect of the robustness of the study, is the geographical scale of the analysis. It's not entirely clear what the scale of a typical postal code area is, although respondents in postal code areas larger than 16 km² were excluded. It would be useful to be clear here on the average size of a postal code area, the minimum, maximum, standard deviation and so on. Following on from that what are the limitations and potential errors of the exposure assessment for environmental metrics? Given that the bird metric is based on the closest eBird locality with a limit of 1.06 km, I'm guessing that the bird metric is more accurate for smaller versus larger postal code areas and therefore in inner city areas versus more peripheral areas? Similarly the NDVI metric based on a 500 metre centroid buffer is likely to be less representative for larger postal code areas? These kinds of issues are discussed a little in limitations, but this could be expanded upon.

We include the following summary of postal code size on line 345:

“In the final data set, mean postal code size was 5.8 km², median was 5.1 km² with a standard deviation of 3.9 km².”

And the following to the discussion on line 298:

“Although we limited the size of postal codes in our analysis, there may be variation in species diversity within each postal code, *differing in accuracy based on the size of the postal code*, influencing confidence intervals.”

3. Results general: Even though the limitations are clearly acknowledged elsewhere, in some places causal language is used i.e. “effect” – relationships should probably be described as associations, as per the paper title.

Thank you for this catch, we have replaced ‘effect’ and ‘relationship’ with ‘association’ throughout.

4. Results Line 103+: it would be useful here to have a really clear statement of the starting n, the exclusions, and the ultimate analysis sample size.

We clarify on Line 120:

(see Materials and Methods, *final sample size for self-rated mental health* n= 47,623 and *self-rated stress* n= 48,693).

5. Results Line 114: I think this should be low self-rated stress?

Yes, thank you for that catch!

6. Results Line 170-172: This is a bit confusing as it describes that distance to blue space became non-significant and then the next sentence suggests distance to blue space had a negative effect? Also “Decreased distance to blue space had a negative effect on self-rated mental health” is not really clear on whether this means living closer to blue space is associated with better or worse mental health – just needs rephrasing.

We deleted this sentence.

7. Results Line 125+ (and elsewhere): I’d hesitate to call this a DAG, which is I think more typically used for causal inference and treats each variable individually. I think this is probably better termed a conceptual model or similar?

We rephrased this as:

“We created a *conceptual model* to explore the relationship between mental health (dependent variables), species diversity and greenspace/bluespace (independent variables), and effect modifiers and potential covariates”

8. Discussion: A key element that is missing for me (also perhaps warranting a mention in the introduction) is a substantive discussion of hypothesised mechanisms. There are a couple of hints in the discussion e.g. evidence on avian diversity and emotional responses, but I'd like to see a more substantive section of the discussion dedicated to theory and evidence on why we might expect tree and bird biodiversity to promote mental health above and beyond more generic 'greenspace'.

We have reordered the paragraph in the introduction to clarify the hypothesized mechanisms of the relationship between species diversity and mental health. Although the mechanisms have been theorized (e.g., the Biophilia Hypothesis and 'restoring capacities' – Marselle et al 2021), how biodiversity leads to better mental health outcomes is not well understood. We now include the following in the introduction:

The restorative effects of biodiversity are thought to relate to human evolution, throughout which our species has relied on a variety of species for survival and reproduction (12). Thus, the 'biophilia hypothesis' posits that humans have an innate affinity to connect with other species and nature(13, although for a critique of the Biophilia Hypothesis see 28). Higher biodiversity may represent a safe environment, where a variety of species mean that our needs are being met, leading to psychological restoration and perceived restorativeness (29). However, how contact with biodiversity leads to better mental health and well-being outcomes is not well understood (30).

In the discussion (line 217) we have added text around Marselle et al's conceptual pathway linking biodiversity to human health:

Three mechanistic pathways by which biodiversity benefits health have been proposed: reducing harm (regulation of air and noise pollution and extreme heat), restoring capacities (restoring psychological and cognitive resources), and building capacities (strengthening of capabilities for meeting everyday demands; 30)... Thus birds have great potential for providing restorative benefits and *building capacities* for humans in cities... Street trees and urban forests are known to provide various ecosystem services for human health and well-being, from air quality to controlling heat (i.e., '*reducing harm*' pathway of biodiversity health benefits, 60). Given their rich cultural resonance, trees feature prominently in urban design and urban forests are valued as symbols comfort, peacefulness, beauty, a connection to nature and biodiversity ('*restoring and building capacity*' through contributing to place identity and attention restoration; 61).

9. Fig 3: Is a useful summary of the main results. I did wonder whether it would be easier to read if variables were grouped together though (i.e. all smoking indicators together, biodiversity/green space all together etc.), I'm not sure having them ordered by OR magnitude is particularly useful? I'm happy to be convinced otherwise though.

The reason we organized them by parameter estimate is so we could zoom in on bird and tree diversity and the other variables that had a similar effect size (e.g., fruit and vegetable consumption). If variables were grouped by type, those with similar effect size would be difficult to find. One of our conclusions is that fruit and vegetable consumption and bird diversity have similar effect sizes, so we would prefer to retain the original grouping.

Reviewer #3 (Remarks to the Author):

In this manuscript, the authors investigate the association between biodiversity in Canadian cities and indicators of mental health.

The introduction is well written. It briefly discusses the main pathways and theories that link biodiversity to health benefits. The objectives are clear. In addition to the main objective of exploring the relationship between biodiversity indicators (trees and birds) and mental health, the authors also include appropriate stratified analyses (by SES). The authors have used appropriate statistical techniques and have properly adjusted models for SES and behavioral background variables, which are known to have an impact on the relationship between green space and health.

The authors found strong evidence for a protective effect of bird diversity on mental health. The evidence for tree diversity was weaker, especially in low SES strata or in models adjusted for all background variables. Nevertheless, the effect size of trees was higher than the effect size of birds. As elsewhere reported, the effect sizes of background variables were larger than those of bird and tree diversity. I particularly liked that the authors could compare the effect size of bird and tree diversity to the effect of daily fruit and vegetable consumption - this is a message that can be picked up easily by media and press, increasing the impact of the manuscript. The risk associated with decreasing distance to blue space is in contrast to earlier findings. Also the association between higher greenness and higher stress is not in line with most literature, unless higher greenness could be seen as an indicator for lower SES in this case.

The discussion is well referenced, follows a logical structure and is well written, offering clear management implications (eg tree planting and avian habitat restoration, represent an opportunity to address both mental health challenges and biodiversity conservation (again, something that can be picked up by media). The authors acknowledge the limitations of the ecological design and the self-reported nature of the health indicators, as well as the potential bias caused by individual variation in nature-relatedness of inhabitants of the studied areas.

Thank you so much for your kind summary of our paper!

29th Apr 24

Dear Dr Buxton,

Your manuscript titled "Mental health is positively associated with biodiversity in Canadian cities" has now been seen by our reviewers, whose comments appear below. In light of their advice we are delighted to say that we are happy, in principle, to publish a suitably revised version in Communications Earth & Environment under the open access CC BY license (Creative Commons Attribution v4.0 International License).

We therefore invite you to revise your paper one last time to address the remaining concerns of our reviewers. At the same time we ask that you edit your manuscript to comply with our format requirements and to maximise the accessibility and therefore the impact of your work.

EDITORIAL REQUESTS:

*****Please take care to match our formatting and policy requirements. We will check revised manuscript and return manuscripts that do not comply. Such requests will lead to delays. *****

SUBMISSION INFORMATION:

OPEN ACCESS:

Communications Earth & Environment is a fully open access journal. Articles are made freely accessible on publication under a CC BY license (Creative Commons Attribution 4.0 International License). This license allows maximum dissemination and re-use of open access materials and is preferred by many research funding bodies.

For further information about article processing charges, open access funding, and advice and support from Nature Research, please visit <https://www.nature.com/commsenv/article-processing-charges>

At acceptance, you will be provided with instructions for completing this CC BY license on behalf of all authors. This grants us the necessary permissions to publish your paper. Additionally, you will be

asked to declare that all required third party permissions have been obtained, and to provide billing information in order to pay the article-processing charge (APC).

[link redacted]

Best regards,

Martina Grecequet, PhD
Associate Editor,
Communications Earth & Environment
@CommsEarth

REVIEWERS' COMMENTS:

Reviewer #1 (Remarks to the Author):

I would like to thank the authors for their careful consideration of all my initial comments. Important additions have been made to the introduction. Moreover, referring to “associations” (rather than “effects”), “covariates” (rather than “confounders”), and “conceptual model” (rather than “DAG”) were all important changes made by the authors. I still don’t think one should compare estimates of exposures and covariates (as models have not been modelled for covariates, and, therefore, estimates for covariates may be biased). However, it is the authors’ decision, and I don’t have any further comments. I congratulate the authors, and wish them all the best for their paper.

Reviewer #2 (Remarks to the Author):

Overall, thanks for a thorough response and revision of the manuscript. I just had one remaining issue, relating to the response to an issue highlighted by reviewer 1. They may well pick this up too:

R1 response – Lines 132-136: I think the response here could be improved a bit. We have growing evidence on nature exposure and health behaviours, especially physical activity. There is also a little evidence on other health behaviours (a quick google scholar search throws up a suggestion or two, including from the CCHS - <https://doi.org/10.1016/j.envres.2022.113124>). So I agree with R1 that health behaviours can reasonably be considered as potential substantive mediators of the relationship between biodiversity exposure and mental health. It is conceivable that inclusion of health behaviours - including total leisure PA, which is likely to be related to outdoor PA - may be diluting associations through mediation.

Reviewer #3 (Remarks to the Author):

No further comments.

We therefore invite you to revise your paper one last time to address the remaining concerns of our reviewers. At the same time we ask that you edit your manuscript to comply with our format requirements and to maximise the accessibility and therefore the impact of your work.

REVIEWERS' COMMENTS:

Reviewer #1 (Remarks to the Author):

I would like to thank the authors for their careful consideration of all my initial comments. Important additions have been made to the introduction. Moreover, referring to “associations” (rather than “effects”), “covariates” (rather than “confounders”), and “conceptual model” (rather than “DAG”) were all important changes made by the authors. I still don’t think one should compare estimates of exposures and covariates (as models have not been modelled for covariates, and, therefore, estimates for covariates may be biased). However, it is the authors’ decision, and I don’t have any further comments. I congratulate the authors, and wish them all the best for their paper.

Thank you for these positive comments!

Reviewer #2 (Remarks to the Author):

Overall, thanks for a thorough response and revision of the manuscript. I just had one remaining issue, relating to the response to an issue highlighted by reviewer 1. They may well pick this up too:

R1 response – Lines 132-136: I think the response here could be improved a bit. We have growing evidence on nature exposure and health behaviours, especially physical activity. There is also a little evidence on other health behaviours (a quick google scholar search throws up a suggestion or two, including from the CCHS - <https://doi.org/10.1016/j.envres.2022.113124>). So I agree with R1 that health behaviours can reasonably be considered as potential substantive mediators of the relationship between biodiversity exposure and mental health. It is conceivable that inclusion of health behaviours - including total leisure PA, which is likely to be related to outdoor PA - may be diluting associations through mediation.

Thank you for this paper. While we agree that there may be a relationship between greenness and health behaviours, such as substance use, there is little evidence of a mechanism between bird and tree species diversity and health behaviours. We found no correlation between greenness and bird or tree diversity in our data. Moreover, it may also hold true that the confounding of health behaviours could work the other way, and that associations could be stronger if health behaviours were better measured (i.e., there could be residual confounding due to imperfect

measurement of all these factors). More work is clearly needed to better understand the confounding and mediating roles of health behaviours on the associations that were observed.

To clarify mediation may be occurring, we added a caveat to the discussion:

“or mediated by health behaviours known to be driven by greenness (e.g., substance abuse; 85)”